# Study on the Use of Artificially Generated Objects in the Process of Training MLP Neural Networks Based on Dispersed Data

**DOI:** 10.3390/e25050703

**Published:** 2023-04-24

**Authors:** Kwabena Frimpong Marfo, Małgorzata Przybyła-Kasperek

**Affiliations:** Institute of Computer Science, University of Silesia, Bȩdzińska 39, 41-200 Sosnowiec, Poland; kwabena.marfo@us.edu.pl

**Keywords:** neural network, multilayer perceptron, artificial training objects, independent data sources, dispersed data

## Abstract

This study concerns dispersed data stored in independent local tables with different sets of attributes. The paper proposes a new method for training a single neural network—a multilayer perceptron based on dispersed data. The idea is to train local models that have identical structures based on local tables; however, due to different sets of conditional attributes present in local tables, it is necessary to generate some artificial objects to train local models. The paper presents a study on the use of varying parameter values in the proposed method of creating artificial objects to train local models. The paper presents an exhaustive comparison in terms of the number of artificial objects generated based on a single original object, the degree of data dispersion, data balancing, and different network structures—the number of neurons in the hidden layer. It was found that for data sets with a large number of objects, a smaller number of artificial objects is optimal. For smaller data sets, a greater number of artificial objects (three or four) produces better results. For large data sets, data balancing and the degree of dispersion have no significant impact on quality of classification. Rather, a greater number of neurons in the hidden layer produces better results (ranging from three to five times the number of neurons in the input layer).

## 1. Introduction

A major problem in the domain of solving problems using machine learning is the decentralization of data sets and the inconsistency of information stored in local independent bases. When data is collected independently by institutions such as banks, hospitals, and various types of mobile applications, one cannot expect the format of the data to be uniform and consistent. Rather, one should expect that different sets of attributes and different sets of objects are present in local tables. Additionally, inconsistencies in data very often occur. The research presented in this paper deals precisely with the issue of classification based on dispersed data. By dispersed data, we mean data that are collected in several decision tables that contain inconsistencies, have different sets of attributes, and objects with the possibility that some attributes and objects may be common among decision tables. In addition, it is almost impossible to identify which objects are common among decision tables since to do that would require the existence of some central identifier of objects, which more often than not does not exist or may not be accessible due to data protection reasons.

The two main approaches that can be used for dispersed data are ensemble of classifiers and federated learning. Ensemble learning is a general approach of creating local models independently based on local tables [1,2], after which a final prediction is generated based on the local models by applying some fusion method [3,4,5]. In this approach, there is no global model as such.

In federated learning, a global model is built which constitutes the main objective presented in [6,7]. In this approach, the main focus is on data protection and data privacy [8]. Here, models are created in local spaces and their parameters only are sent to a central server—local data are not exchanged or combined among local spaces. The local models are then aggregated and sent to the local spaces. Such a procedure is iterated until a convergence criterion is satisfied.

The approach proposed in this paper is quite different. The aim of the method is to build a global model but in a completely different way than in federated learning. Indeed, local models are built based on local tables which are later used to construct a global model; however, this procedure is not iterative. Creation of a global model is carried out by a one-time aggregation. In the final stage, the global model is trained with a stratified subset of the original test set for which the values on the full set of conditional attributes present in all local tables are defined.

In this study, neural networks—multilayer perceptrons (MLP)—are used as local models. For the aggregation of such local networks to be possible, all of them must have the same structure. Since there are different conditional attributes in each local table, obtaining the same input layer in all models is not trivial. It is necessary to artificially generate objects based on the original objects that are to be used to train the network. Such artificial objects must have defined values on the conditional attributes that are missing in the considered local table. The paper proposes a method for generating artificial objects and contains a study on the use of different parameter values in the proposed method of generating artificial objects. An exhaustive comparison in terms of the number of artificial objects generated based on a single original object, the degree of data dispersion, data balancing, and different network structures—the number of neurons in the hidden layer are presented. The main conclusions reached are as follows: it was found that for data sets with a large number of objects, a smaller number of artificial objects is optimal. For smaller data sets, a greater number of artificial objects (three or four) produces better results. For large data sets, data balancing and the degree of dispersion have no significant impact on the quality of classification. Rather, a greater number of neurons in the hidden layer produces better results (ranging from three to five times the number of neurons in the input layer).

The contribution of the paper are as follows:Proposing a method for generating artificial objects for training local MLP networks with identical structure;Comparison of the proposed method in relation to different number of artificially objects generated;Comparison of the proposed method in relation to different versions of data dispersion;Comparison of the proposed method in relation to different number of neurons in the hidden layer;Comparison of the proposed method for balanced and imbalanced versions of data sets.

Neural networks have been considered for dispersed data in various applications. The papers [9,10] considered neural networks as a model for aggregating prediction vectors generated by local classifiers. In the paper [11], neural networks were used in a federated learning approach. Neural networks were also used as base models in an ensemble of classifiers whose predictions were then aggregated by various fusion methods [12]. However, none of the approaches described above is similar to the one proposed in this study. The main difference lie in the non-iterative approach when building the global model in the proposed approach and the use of local tables with different sets of conditional attributes to train local networks with identical structures.

The paper is organized as follows. In Section 2, the proposed method for generating a global model is described. The section explains how to determine the structure of local models and how to prepare artificial objects for training local models. Then, the method of aggregating local models to the global model and the stage of training the global model are described. Section 3 addresses the data sets that were used and presents the conducted experiments, comparisons, and discussion on obtained results. Section 4 is on conclusions and future research plans.

## 2. Materials and Methods

The main idea of the proposed model is to build a global model based on dispersed data—local tables with different sets of conditional attributes—in three stages:First stage: training local models, MLP neural networks based on local tables;Second stage: aggregation of local models to the global model. This stage is performed in a non-iterative way by a single calculation;Third stage: post-training the global model using a stratified subset of the original test set.

All three stages are described below in separate subsections.

### 2.1. First Stage—Training Local Models, MLP Neural Networks, Based on Local Tables

Formally, dispersed data is a set of decision tables that are collected independently by separate units. We assume that a set of decision tables—local tables $D_{i}=\left(U_{i},A_{i},d\right)\;i\in \left\{1,\ldots ,n\right\}$ from one discipline—is available, where $U_{i}$ is the universe comprising a set of objects; $A_{i}$ is a set of conditional attributes; and *d* is a decision attribute. We assume that the sets of conditional attributes of local tables are quite different although it may rarely happen that a larger set of attributes is common between tables. More likely, the differences in attributes found in local tables are significant.

The local models that are used in this study are multilayer perceptron networks (MLP). Based on each local table, an MLP model is trained separately. The desired objective that all local models must have the same structure is not trivial since each local table has different conditional attributes, thus making the training process difficult. We propose that the input layer of local networks contains all the attributes that are present in all local tables—let us denote this set as $A={⋃}_{i\in \{1,\ldots ,n\}}\;A_{i}$. In addition, the hidden layer should contain the same number of neurons in all networks. The output layer will be same for all tables due to the identical decision attribute present in all local tables. In this study, we use only one hidden layer in the network.

Now, a problem arises when we seek to train such a network based on a single local table given that the table in question lacks conditional attributes (perhaps many) that are present in the input layer of the network. A method for generating artificial objects with supplemented values on missing conditional attributes is proposed. These values are imputed based on certain characteristics provided by other local tables in the dispersed data in which the missing attributes are present. In doing so, data protection is ensured because we do not exchange raw data but only certain values of statistical measures derived from the dispersed data.

Based on each original object from a local table, *k* artificial objects are generated as follows:Let us consider an object *x* that belongs to a decision class *v* from a local table $D_{i}$.We define a set of tuples as
$$\begin{array}{cc} METHODS & =(min,min),(min,mean)\cdots (max,median),(max,max)\\ \; & \in (min,mean,median,max)\times (min,mean,median,max) \end{array}$$For each missing attribute (attribute from the set $A\backslash A_{i}$) and each method∈METHODS, method(0) is computed on the objects having the decision class *v* for all local tables in which the attribute is present. After, method(1) is computed on the the resulting values from method(0).After step 2, there will be |METHODS|=16 values for decision class *v*. *k* distinct values denoted by $a_{k}$ are randomly selected from the 16 values, where *k* is the number of artificial objects that are to be generate.From step 3, there will be *k* derived values for all the missing attributes of object *x*.The final step is to duplicate object *x*, *k* times, and assign the $a_{k}$ values to the missing attribute.

This process is carried out for all objects in a local table and executed separately for each local table.

A training set of artificially prepared objects as described above is then used to train the MLP network. The neural networks is implemented using the Keras library in Python. Different number of neurons in the hidden layer is experimented on—values ranging from 0.25 to 5 times the number of neurons from the input layer are tested. For the hidden layer, the ReLU (Rectified Linear Unit) activation function is used as it is the most popular activation function and gives very good results [13]. For the output layer, the Softmax activation function is used, which is recommended when we deal with a multi-class problem [14]. The neural network is trained by using a gradient descent method with an adaptive step size in the backpropagation method. The Adam optimizer [15] and the categorical cross-entropy loss function [16] are used in the study.

### 2.2. Second Stage—Aggregation of Local Models to the Global Model

The second stage consists of aggregation of local networks into a single global network. In the first stage, the local neural networks are prepared in such a way that aggregation is possible—all local networks have the same structure; thus, the global network will also have the same network structure. The weights in global model are determined based on the weighted average of the corresponding weights from the local models. However, due to the dispersed data stored in the local tables, not all local models are equally accurate, so the weighted average is employed to make the local model’s influence on the construction of the global model depend on the accuracy of a given local model. The method used is inspired by the second weighting system used in the AdaBoost algorithm [17].

For each local model, a classification error is estimated based on its training set (containing artificial objects). Let us denote by $e_{i}$ the classification error determined for the *i*-th local model i∈{1,…,n}. Since local models are built based on a piece of data, their accuracy can be very different. It may sometimes happen that their classification error is above 0.5. In order not to eliminate such local models from the aggregation stage as they may contain important information on specific attributes that may have a positive impact in the global model, the min-max normalization is applied to the interval [0,0.5] of all errors $e_{i},i\in \left\{1,\ldots ,n\right\}$. After, the weights $\omega_{i}$ for each local neural network i∈{1,…,n} are adjusted according to the formula proposed in [17]:(1)$$\omega_{i}=ln\left(\frac{1-e_{i}}{e_{i}}\right)$$
The initial weights of the global model between neural connections are then calculated based on the adjusted weights of all the local networks. More specifically, the weights of the global model are determined by the weighted average of adjusted weights $\omega_{i},i\in \left\{1,\ldots ,n\right\}$.

It should be noted that some attributes that appear more frequently in local tables may have been better trained in global model than others. Therefore, a MLP network created in this way does not always generate sufficiently good results. In the next stage, the quality of the network is improved.

### 2.3. Third Stage—Post-Training the Global Model Using a Small Training Set

In order to implement this step, it is necessary to have access to an independent set of training data which can be called a global training set. This means that each object in this set has values for all conditional attributes *A* from the dispersed data. This set cannot be generated from local tables since aggregation is not possible considering the assumptions about dispersed data mentioned earlier.

Such a global training set is extracted from the test set. The test set is divided into two equal parts in a stratified manner. One is used for the post-training stage and the other for testing. This procedure is repeated twice where each time a different half is used for the post-training phase. In future studies, it is planned to generate such a global training set artificially.

## 3. Results

The experiments are conducted with data taken from the UC Irvine Machine Learning Repository. Three data sets are selected: Vehicle data [18], Landsat Satellite data [19], and Dry Bean data [20]. Each data set available in the repository is stored in a single table. These data sets are chosen for three reasons. To begin, these data sets are chosen because of the presence of multiple decision classes in the sets as the proposed method is tested for multi-class problems. Additionally, an important factor is the significant number of conditional attributes present in the data sets. The data are dispersed into local tables in the way where the conditional attributes are split. The aim is to test the approach where we have different conditional attributes in local tables. To achieve this, a large number of attributes is needed originally so that such dispersion can occur and a meaningful subset of these attributes can be present in each local table. Lastly, in this study, we focus on using numerical data—there are numerical, discrete, or continuous attributes in all data sets. Due to the large variation in the attributes in the Dry Bean data, the set is normalized.

The only possible way to evaluate the model for the considered dispersed data is the train-and-test method. This is because the data in the local tables contain only subsets of conditional attributes, while we assume that the test objects will already have specified values for all possible attributes present in the local tables. So, before the original data set is dispersed, it is divided into a training set (70% of objects) and a test set (30% of objects) in a stratified manner. Data characteristics are given in Table 1. The training data sets are then dispersed into local tables. Different degrees of dispersion are considered in order to check whether the method can cope with significant data dispersion. The creation of versions with 3,5,7,9, and 11 local tables based on the original training set are considered where all local tables contained only a subset of the original set of conditional attributes. In addition, different local tables had different sets of attributes; however, there is a possibility of individual attributes being present among some tables. The decision attribute is included in each of the tables. The full set of objects is also stored in each of the local tables but without identifiers. This reflects the real situation where one cannot identify the objects between local tables.

All the data sets are heavily imbalanced Figure 1. To check whether the proposed method can handle imbalanced data, each data set is considered in two versions—the imbalanced version and the balanced version. The data are balanced with the use of the synthetic minority over-sampling technique (SMOTE) method [21]. The implementation of this algorithm, available in WEKA [22] software, is used. The balancing procedure is performed for each local table separately using only the locally available subset of attributes. All objects for each decision class are balanced in a way that after the implementation of this process, each decision class has an equal number of objects as the decision class with the most objects in the set. Thus, a total of thirty dispersed sets are analyzed: each of the three data sets is dispersed into 5 versions, each version is balanced to a total of 3×5×2.

The quality of classification is evaluated based on the test set. The accuracy measure acc is analyzed. This is the defined as a fraction of correctly classified objects to all objects in the test set.

The main goal of the experiments is to investigate how the number of objects artificially generated based on a single object from a local table affects the quality of classification. An additional purpose is to determine the guidelines that should be followed in determining such an optimal value depending on the characteristics of the data sets as well as to check the effect of the degree of dispersion on the obtained quality of classification. The different network structures and the impact of the number of neurons in the hidden layers on the quality of classification are also studied. Comparison analysis to determine whether the proposed approach performs equally well for balanced and imbalanced data is carried out. To meet these objectives above, the scheme of the experiments is as follows.

Studying different number of artificial objects generated based on a single object from each local table. The number of artificial objects generated k∈{1,2,3,4,5} are studied.Studying different levels of dispersion: 3, 5, 7, 9, 11 local tables.Studying different number of neurons in the hidden layer. The number is determined in proportion to the number of neurons in the input layer. The following values are tested: {0.25, 0.5, 0.75, 1, 1.5, 1.75, 2, 2.5, 2.75, 3, 3.5, 3.75, 4, 4.5, 4.75, 5} × the number of neurons in the input layer.Studying two versions for each data set—balanced and imbalanced versions.Studying an iterative approach modeled on federated learning in order to make comparisons with the proposed approach.

Comparison of experimental results is made in terms of:The quality of classification for different number of artificial objects generated;The quality of classification for different versions of dispersion;The quality of classification for different number of neurons in the hidden layer;The quality of classification for balanced and imbalanced version of data sets.

Table A1, Table A2, Table A3, Table A4, Table A5 and Table A6, presented in Appendix A, show the classification accuracy obtained for different versions of dispersion, different numbers of artificially generated objects, and different numbers of neurons in the hidden layer for Vehicle imbalanced, Vehicle balanced, Landsat Satellite imbalanced, Landsat Satellite balanced, Dry Bean imbalanced and Dry Bean balanced data sets. Each experiment is performed three times. The average of the three runs is given in the tables below. In each row of the tables, the best result is in a bold font. The following sections present an analysis of the results included in these tables from different perspectives. The last part presents a comparison with the approach modeled on federated learning.

### 3.1. Comparison of Quality of Classification for Different Numbers of Objects Artificially Generated

First, we compare the quality of classification using different number of artificially generated objects. Table 2 shows a comparison of the best results (those in a bold font in Table A1, Table A2, Table A3, Table A4, Table A5 and Table A6) obtained for different number of artificially generated objects. In the table, for each dispersed data set, the best result is shown in a bold font.

As can be seen, for different data sets, different numbers of artificially generated objects guarantee the best results. In the case of the Vehicle data set, it can only be said that the approach with one artificial object gives the worst results. In the case of the Dry Bean data set, definitely the use of two artificial objects generates the best results. For the Landsat Satellite data set, it is hard to define any of these types of relations.

Statistical tests are performed in order to check the importance in the differences in the obtained results acc for different number of objects artificially generated. The Friedman’s test using all results from Table 2 is performed. Five dependent groups are analyzed ({1,2,3,4,5} number of artificial objects). The test did not confirm that differences among the classification accuracy in these five groups are significant (p=0.672). However, as can be seen from Table 2, the classification accuracy obtained for different data sets are from completely different ranges. Due to this discrepancy, it is difficult to prove the significance of the differences. Therefore, it was decided to separate the obtained results against the considered data sets. Thus, three sets (for Vehicle, for Landsat Satellite, and for Dry Bean) each containing a ten-element sample are obtained. The Friedman’s test confirmed the significance of the differences for the Dry Bean data set with p=0.003. The Wilcoxon each-pair test confirmed the significant differences between the average accuracy values for the following pairs: Vehicle—2 and 4 artificial objects, p=0.01; Landsat Satellite—1 and 3 artificial objects, p=0.03; Dry Bean—2 and 1 artificial objects, p=0.008, 2 and 3 artificial objects, p=0.006, 2 and 4 artificial objects, p=0.008, 2 and 5 artificial objects, p=0.004.

Additionally, comparative box-plot charts for the values of the classification accuracy and different data sets are created (Figure 2). As can be observed, for the Dry Bean data set, the box-plot for the two artificial objects definitely stands out among the others. It can also be concluded that using a single artificial object never generates good results. Taking into account the results of the comparisons and the number of objects in the analyzed data sets, a general conclusion can be drawn. For data sets with a large number of objects (around 9000 objects), a smaller number of artificial objects such as two objects is optimal. For smaller data sets with up to a thousand objects, a greater number of artificial objects (three or four) produces better results. More specifically, the smaller the number of objects in the local tables, the more artificially generated objects should be used in the proposed approach.

### 3.2. Comparison of Quality of Classification for Different Versions of Dispersion

We now compare the classification accuracy obtained for different versions of data dispersion. In Table 3 a comparison of the best results (those bolded in Table A1, Table A2, Table A3, Table A4, Table A5 and Table A6) obtained for different version of dispersion is presented. In the table, for data set, the best result is shown in a bold font.

As can be observed, in the case of Vehicle data set, the best results are obtained for medium data dispersion (7 local tables) or even large data dispersion (11 local tables). For this data set, the differences in results obtained for different versions of dispersion are the greatest compared to the other data sets. For Landsat Satellite and Dry Bean data sets, the smallest dispersion (3 local tables) gives better results. However, looking closely at the results, we can observe that the absolute differences noted for these data sets are really small—at the third decimal place. So, we can conclude that for data sets with such a large number of objects, the differences recorded for different degrees of dispersion are really unremarkable.

Statistical tests are performed in order to confirm the importance in the differences in the obtained results acc. At first, the values of the classification accuracy in five dependent groups (3,5,7,9,11 local tables) are analyzed. The Friedman test confirmed a statistically significant difference in the results obtained for the five different version of dispersion being considered, $\chi^{2}\left(28,4\right)=26.608,p=0.00003$. The Wilcoxon each-pair test confirmed the significant differences between the average accuracy values for all pairs with 11 local tables: 3 and 11 local tables p=0.007, 5 and 11 local tables p=0.001, 7 and 11 local tables p=0.004, 9 and 11 local tables p=0.016.

Additionally, a comparative box-plot chart for the values of the classification accuracy is created (Figure 3). Here, the distributions of the values obtained for different versions of dispersion are similar; thus, we can conclude that for sufficiently large data sets (5000 objects), the degree of dispersion does not have a huge impact on the obtained results. More specifically, the degree of dispersion has little effect on the quality of classification in the proposed approach.

### 3.3. Comparison of Quality of Classification for Different Numbers of Neurons in the Hidden Layer

In Table A1, Table A2, Table A3, Table A4, Table A5 and Table A6, which are presented earlier, all the results obtained for the different analyzed number of neurons in the hidden layer are given. The best obtained classification accuracies are also marked in those tables. It can be seen that these best results are generated by a higher number of neurons in the hidden layer. The optimal values are above 3× the number of neurons in the input layer up to 5× the number of neurons in the input layer. This propriety does not depend on the number of objects in data set—no matter how large the data set is, more neurons in the hidden layer gives better results. However, there is not one universal number of neurons in the hidden layer that is optimal for every data set.

In order to notice certain patterns for particular data sets, heat maps are created based on the results from Table A1, Table A2, Table A3, Table A4, Table A5 and Table A6 and shown in Figure 4. On the *x*-axis, the number of neurons in the hidden layer is presented, while the number of artificial objects generated and the version of the dispersion are shown on the *y*-axis. The color on the map is determined by the classification accuracy value. Definitely for the Dry Bean data set, the clearest pattern can be seen, which shows that increasing the number of neurons in the hidden layer clearly improves classification accuracy. Additionally, for the Vehicle data set, it can be seen that a higher number of neurons results in better quality. The least visible dependence is found in the heat map for the Landsat Satellite data set. Here, for a large number of neurons in the hidden layer, both very good classification quality and worse results were observed. More specifically, it depends on the data set whether the increased number of neurons in the hidden layer will improve the quality of classification, and this impact is very different and specific to the data set.

### 3.4. Comparison of Quality of Classification for Balanced and Imbalanced Versions of Data Set

We will now focus on comparing the results obtained for balanced and imbalanced data. In Table 4, a comparison of the best results (those in a bold font in Table A1, Table A2, Table A3, Table A4, Table A5 and Table A6) obtained for balanced and imbalanced versions of each dispersed data is presented. In the table, the best result is shown in a bold font for each dispersed data set.

Based on the results, it cannot be explicitly concluded that the proposed method gives better results for balanced only or imbalanced data only as it depends on the data set in question. For the Vehicle data set, better results are obtained with balanced data, while for the Landsat Satellite data set, better results are obtained with imbalanced data. In both cases, the Wilcoxon test for dependent samples confirmed the statistical significance of the differences with p=0.0001. In contrast, for the Dry Bean data set, the results in both balanced and imbalanced versions are virtually the same. Here, the Wilcoxon test did not confirm the significance of the differences (p=0.523).

Comparative box-plot charts for the values of the classification accuracy in two groups of imbalanced and balanced data are created (Figure 5). The graphs confirm earlier conclusions; hence, it can be said that the proposed method handles balanced and imbalanced data comparably. In fact, the final result depends on the specifics of the data set. Determining the specific characteristics of the data sets that influenced the results requires further study. More specifically, it depends on the data set whether applying the SMOTE method for balancing the data set improves the quality of classification.

### 3.5. Comparison of Quality of Classification of the Proposed Approach with an Iterative Approach Modeled on Federated Learning

In this section, the results obtained from the approach modeled on federated learning [7,8,11] will be presented. Then a comparison will be made with the results obtained for the proposed approach.

The main difference between the proposed approach and the one based on federated learning is the iterative aggregation of local models. In the proposed approach, local models aggregation occurs only once. The approach modeled on federated learning involves the following steps:Generation of local MLP neural networks based on local tables created analogously as described in Section 2.1. This means that missing attributes are filled in local tables, and artificial objects are generated.The obtained weights and biases from local models are sent to a central server.At the central server, the average of the weights and biases are computed, and the global model obtained is sent back to the local devices.Local devices accept the global model, and once again, trained weights and biases are sent to the central server. Steps 3 and 4 are iterated three times.The global model is post-training on a stratified half of the test set and its accuracy is tested on the remaining half. At another step, the global model is post-training on the other half and tested on the remaining half, after which the classification accuracy is averaged. This is the final step of the process.

As may be noted, an effort was made to provide a fair comparison as both the artificial objects and the post-training process were used in the above approach. An important difference between the proposed approach and the above model is the iterative aggregation of the global model. In addition, the same numbers of artificial objects generated and the same number of neurons in the hidden layer were also analyzed. Of course, the experiments were performed on the same data sets in terms of the degree of dispersion and balanced/imbalanced version. The full results are not given here for the sake of readability and clarity of the paper. Table 5 gives comparison of the results obtained for the proposed approach and the one based on federated learning. In the table, a better result from the two approaches is shown in a bold font. As can be seen, in the overwhelming number of cases, the proposed approach produced better results. Only in thirteen cases for the Vehicle data set did the approach modeled on federated learning produce slightly better results.

Statistical tests are performed to confirm the significance of the differences in the obtained results acc for the proposed approach and the approach modeled on federated learning. The Wilcoxon test using all results from Table 5 is performed. Two dependent groups are analyzed (PA—the proposed approach, FL—the approach modeled on federated learning). The test confirms that differences among the classification accuracy in these two groups are significant (p=0.0001). Additionally, comparative box-plot charts for the values of the classification accuracy are created (Figure 6). The graphs confirm earlier conclusions, and hence it can be said that the proposed method generates better results than the approach modeled on federated learning.

## 4. Conclusions

The paper presented a new method for generating a global MLP model based on dispersed data with different sets of conditional attributes present in local tables. The novelty proposed is the method of generating artificial objects to train local networks with identical structure. An exhaustive comparison of the proposed method has been carried out in terms of the number of artificially generated objects, network structure, data balancing, and degree of data sparseness. The main conclusions are as follows. The greater the number of objects in local tables, the smaller the number of artificially generated objects is sufficient to generate optimal results. For smaller data sets, a greater number of artificial objects (three or four) produces better results. For large data sets, data balancing and the degree of dispersion have no significant impact on the quality of classification. In most cases, a higher number of neurons in the hidden layer gives better results; however, this is very data-dependent and specific. The best results are obtained for the number of neurons in the hidden layer equal to three to five times the number of neurons in the input layer. The paper also confirmed that the proposed method gives better results than the method modeled on federated learning.

In the proposed approach, many aspects should be considered in the future. Among the main plans are to test other ways of aggregating local models and proposing a new method for generating a global training set used in the post-training phase.

## Figures and Tables

**Figure 1 entropy-25-00703-f001:**
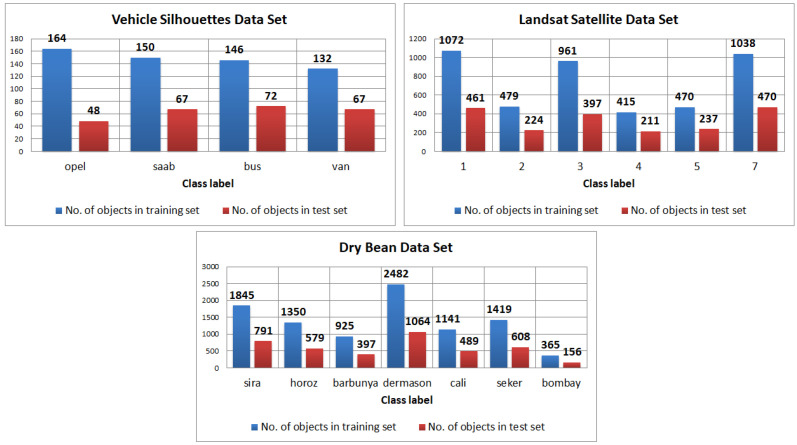
Imbalance of data—cardinality of decision classes in training and test sets.

**Figure 2 entropy-25-00703-f002:**
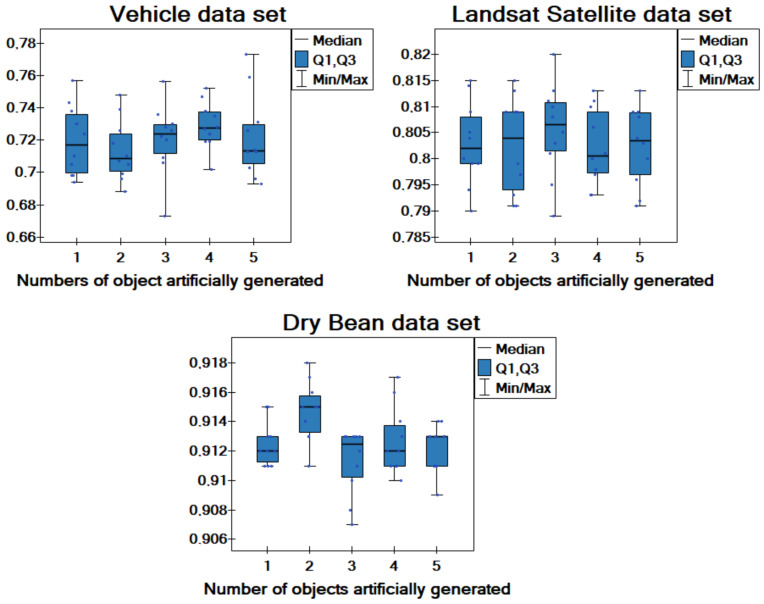
Box-plot chart with (median, the first quartile—Q1, the third quartile—Q3) the value of classification accuracy acc for the different numbers of objects artificially generated.

**Figure 3 entropy-25-00703-f003:**
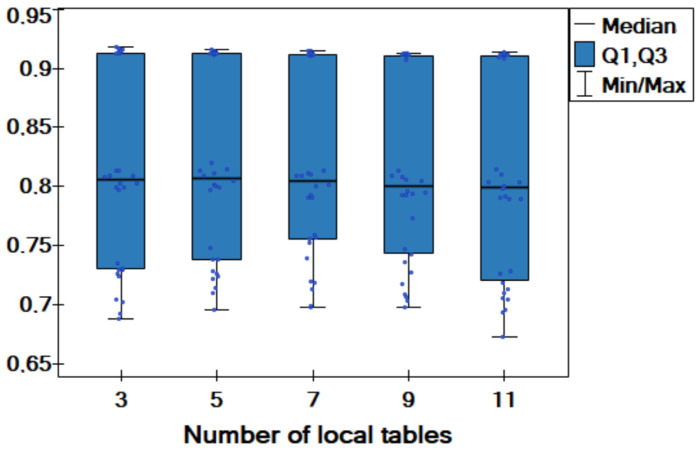
Box-plot chart with (median, the first quartile—Q1, the third quartile—Q3) the value of classification accuracy acc for different versions of dispersion.

**Figure 4 entropy-25-00703-f004:**
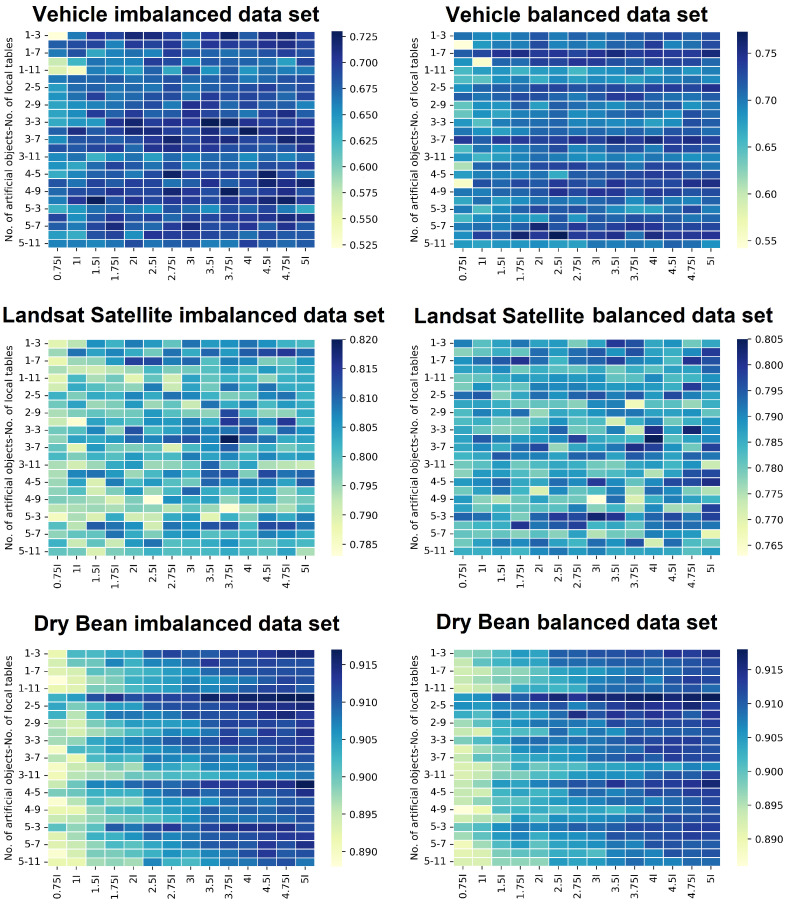
Heat maps on the accuracy levels of all data sets.

**Figure 5 entropy-25-00703-f005:**
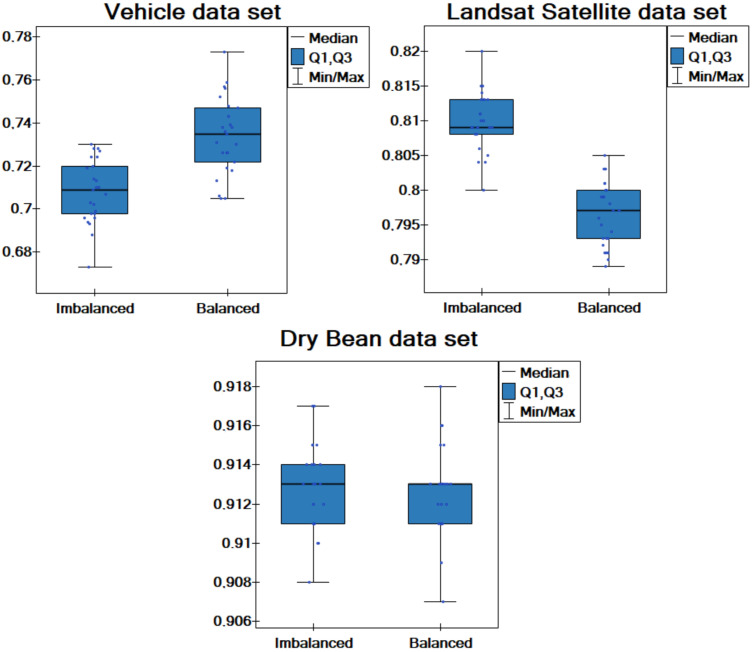
Box-plot chart with (median, the first quartile—Q1, the third quartile—Q3) the value of classification accuracy acc for imbalanced and balanced versions of data sets.

**Figure 6 entropy-25-00703-f006:**
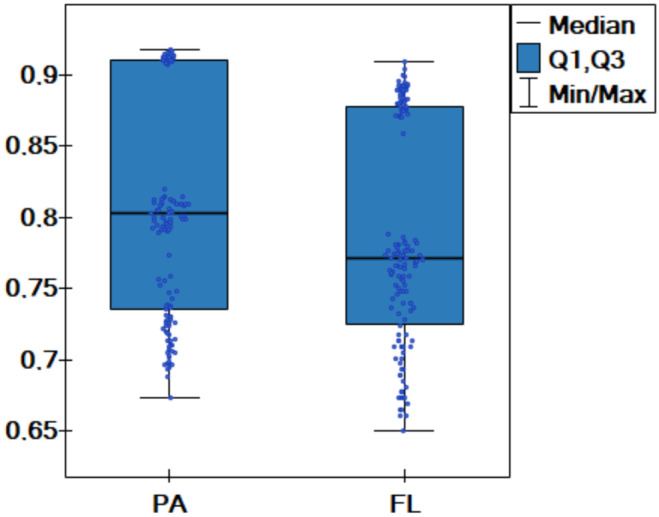
Box-plot chart with (median, the first quartile—Q1, the third quartile—Q3) the value of classification accuracy acc for the proposed approach and the approach modeled on federated learning.

**Table 1 entropy-25-00703-t001:** Data set characteristics. Sign # denotes the number of objects in the set.

Data Set	# The Training Set	# The Test Set	# Conditional Attributes	# Decision Classes
Vehicle	592	254	18	4
Landsat Satellite	4435	1000	36	6
Dry Bean	9527	4084	16	7

**Table 2 entropy-25-00703-t002:** Comparison of classification accuracy acc obtained for different number of artificially generated objects.

Data	No. of	No. of Artificially Generated Objects				
Set	Tables	1	2	3	4	5
Vehicle imbalanced	3	0.724	0.688	**0.73**	0.702	0.693
	5	0.71	0.696	**0.728**	0.724	0.714
	7	0.698	0.699	**0.72**	0.719	0.713
	9	0.698	0.707	0.709	**0.727**	0.703
	11	0.694	0.71	0.673	**0.728**	0.696
Vehicle balanced	3	0.73	0.705	0.726	**0.735**	0.731
	5	0.738	**0.748**	0.722	0.738	0.726
	7	0.757	0.739	0.756	0.752	**0.759**
	9	0.743	0.718	0.736	0.747	**0.773**
	11	0.705	**0.726**	0.706	0.719	0.713
Landsat Satellite imbalanced	3	0.809	0.809	**0.813**	**0.813**	0.808
	5	0.815	0.809	**0.82**	0.811	0.813
	7	**0.814**	0.809	0.811	0.81	0.809
	9	0.805	**0.813**	0.808	0.806	0.809
	11	0.804	**0.815**	0.81	0.8	0.804
Landsat Satellite balanced	3	0.799	0.799	**0.803**	0.797	**0.803**
	5	0.799	0.797	**0.805**	0.801	0.8
	7	0.8	0.791	**0.801**	0.793	0.791
	9	0.794	0.793	0.795	0.793	**0.796**
	11	0.79	0.791	0.789	**0.798**	0.792
Dry Bean imbalanced	3	0.915	**0.917**	0.913	**0.917**	0.914
	5	0.913	**0.915**	0.912	0.914	0.913
	7	0.911	**0.915**	0.911	0.91	0.914
	9	0.912	**0.913**	0.91	0.911	**0.913**
	11	0.912	**0.914**	0.908	0.911	0.911
Dry Bean balanced	3	0.915	**0.918**	0.913	0.916	0.913
	5	0.912	**0.916**	0.913	0.913	0.913
	7	0.911	**0.915**	0.913	0.912	0.911
	9	**0.913**	**0.913**	0.907	0.911	0.911
	11	0.911	0.911	**0.913**	0.912	0.909

**Table 3 entropy-25-00703-t003:** Comparison of classification accuracy acc obtained for different numbers of artificially generated objects.

Data	No. of Artificially	No. of Local Tables				
Set	Generated Objects	3	5	7	9	11
Vehicle imbalanced	1	**0.724**	0.71	0.698	0.698	0.694
	2	0.688	0.696	0.699	0.707	**0.71**
	3	**0.73**	0.728	0.72	0.709	0.673
	4	0.702	0.724	0.719	0.727	**0.728**
	5	0.693	**0.714**	0.713	0.703	0.696
Vehicle balanced	1	0.73	0.738	**0.757**	0.743	0.705
	2	0.705	**0.748**	0.739	0.718	0.726
	3	0.726	0.722	**0.756**	0.736	0.706
	4	0.735	0.738	**0.752**	0.747	0.719
	5	0.731	0.726	0.759	**0.773**	0.713
Landsat Satellite imbalanced	1	0.809	**0.815**	0.814	0.805	0.804
	2	0.809	0.809	0.809	0.813	**0.815**
	3	0.813	**0.82**	0.811	0.808	0.81
	4	**0.813**	0.811	0.81	0.806	0.8
	5	0.808	**0.813**	0.809	0.809	0.804
Landsat Satellite balanced	1	0.799	0.799	**0.8**	0.794	0.79
	2	**0.799**	0.797	0.791	0.793	0.791
	3	0.803	**0.805**	0.801	0.795	0.789
	4	0.797	**0.801**	0.793	0.793	0.798
	5	**0.803**	0.8	0.791	0.796	0.792
Dry Bean imbalanced	1	**0.915**	0.913	0.911	0.912	0.912
	2	**0.917**	0.915	0.915	0.913	0.914
	3	**0.913**	0.912	0.911	0.91	0.908
	4	**0.917**	0.914	0.91	0.911	0.911
	5	**0.914**	0.913	**0.914**	0.913	0.911
Dry Bean balanced	1	**0.915**	0.912	0.911	0.913	0.911
	2	**0.918**	0.916	0.915	0.913	0.911
	3	**0.913**	**0.913**	**0.913**	0.907	**0.913**
	4	**0.916**	0.913	0.912	0.911	0.912
	5	**0.913**	**0.913**	0.911	0.911	0.909

**Table 4 entropy-25-00703-t004:** Comparison of classification accuracy acc obtained for imbalanced and balanced versions of data.

Data Set	No. of Tables	No. of Art. Objects	Imbalanced	Balanced	Data Set	Imbalanced	Balanced
		1	0.724	**0.73**		0.915	0.915
		2	0.688	**0.705**		0.917	**0.918**
	3	3	**0.73**	0.726		0.913	0.913
		4	0.702	**0.735**		**0.917**	0.916
		5	0.693	**0.731**		**0.914**	0.913
		1	0.71	**0.738**		**0.913**	0.912
		2	0.696	**0.748**		0.915	**0.916**
	5	3	**0.728**	0.722		0.912	**0.913**
		4	0.724	**0.738**		**0.914**	0.913
		5	0.714	**0.726**		0.913	0.913
		1	0.698	**0.757**		0.911	0.911
		2	0.699	**0.739**		0.915	0.915
Vehicle	7	3	0.72	**0.756**	Dry	0.911	**0.913**
		4	0.719	**0.752**	Bean	0.91	**0.912**
		5	0.713	**0.759**		**0.914**	0.911
		1	0.698	**0.743**		0.912	**0.913**
		2	0.707	**0.718**		0.913	0.913
	9	3	0.709	**0.736**		**0.91**	0.907
		4	0.727	**0.747**		0.911	0.911
		5	0.703	**0.773**		**0.913**	0.911
		1	0.694	**0.705**		**0.912**	0.911
		2	0.71	**0.726**		**0.914**	0.911
	11	3	0.673	**0.706**		0.908	**0.913**
		4	**0.728**	0.719		0.911	**0.912**
		5	0.696	**0.713**		**0.911**	0.909
		1	**0.809**	0.799			
		2	**0.809**	0.799			
	3	3	**0.813**	0.803			
		4	**0.813**	0.797			
		5	**0.808**	0.803			
		1	**0.815**	0.799			
		2	**0.809**	0.797			
	5	3	**0.82**	0.805			
		4	**0.811**	0.801			
		5	**0.813**	0.8			
		1	**0.814**	0.8			
		2	**0.809**	0.791			
Landsat	7	3	**0.811**	0.801			
Satellite		4	**0.81**	0.793			
		5	**0.809**	0.791			
		1	**0.805**	0.794			
		2	**0.813**	0.793			
	9	3	**0.808**	0.795			
		4	**0.806**	0.793			
		5	**0.809**	0.796			
		1	**0.804**	0.79			
		2	**0.815**	0.791			
	11	3	**0.81**	0.789			
		4	**0.8**	0.798			
		5	**0.804**	0.792			

**Table 5 entropy-25-00703-t005:** Comparison of classification accuracy acc obtained for the proposed approach (PA) and the approach modeled on federated learning (FL).

Approach		PA	FL	PA	FL	PA	FL	PA	FL	PA	FL
**Data**	**No. of**	**No. of Artificially Generated Objects**									
**Set**	**Tables**	**1**	**1**	**2**	**2**	**3**	**3**	**4**	**4**	**5**	**5**
Vehicle imbalanced	3	**0.724**	0.677	**0.688**	0.677	**0.73**	0.673	0.702	**0.709**	0.693	**0.724**
	5	**0.71**	0.681	**0.696**	0.673	**0.728**	0.693	**0.724**	0.717	**0.714**	0.709
	7	0.698	**0.705**	**0.699**	0.693	**0.72**	0.661	**0.719**	0.665	**0.713**	0.701
	9	**0.698**	0.673	**0.707**	0.685	**0.709**	**0.709**	**0.727**	0.697	**0.703**	0.677
	11	**0.694**	0.673	**0.71**	0.673	**0.673**	**0.673**	**0.728**	0.689	**0.696**	0.661
Vehicle balanced	3	**0.73**	0.65	0.705	**0.713**	0.726	**0.752**	**0.735**	0.713	**0.731**	0.689
	5	**0.738**	0.709	**0.748**	0.669	**0.722**	0.701	**0.738**	0.732	**0.726**	0.713
	7	**0.757**	0.709	0.739	**0.748**	**0.756**	0.665	**0.752**	0.736	0.759	**0.764**
	9	**0.743**	0.717	0.718	**0.756**	**0.736**	0.728	0.747	**0.748**	**0.773**	0.748
	11	0.705	**0.709**	0.726	**0.736**	0.706	**0.74**	**0.719**	0.693	0.713	**0.748**
Landsat Satellite imbalanced	3	**0.809**	0.759	**0.809**	0.766	**0.813**	0.773	**0.813**	0.783	**0.808**	0.781
	5	**0.815**	0.766	**0.809**	0.768	**0.82**	0.781	**0.811**	0.78	**0.813**	0.781
	7	**0.814**	0.779	**0.809**	0.77	**0.811**	0.777	**0.81**	0.777	**0.809**	0.769
	9	**0.805**	0.771	**0.813**	0.767	**0.808**	0.784	**0.806**	0.786	**0.809**	0.782
	11	**0.804**	0.771	**0.815**	0.773	**0.81**	0.775	**0.8**	0.781	**0.804**	0.782
Landsat Satellite balanced	3	**0.799**	0.74	**0.799**	0.734	**0.803**	0.743	**0.797**	0.77	**0.803**	0.773
	5	**0.799**	0.74	**0.797**	0.759	**0.805**	0.746	**0.801**	0.777	**0.8**	0.774
	7	**0.8**	0.76	**0.791**	0.752	**0.801**	0.766	**0.793**	0.777	**0.791**	0.773
	9	**0.794**	0.75	**0.793**	0.759	**0.795**	0.762	**0.793**	0.774	**0.796**	0.765
	11	**0.79**	0.757	**0.791**	0.776	**0.789**	0.761	**0.798**	0.763	**0.792**	0.788
Dry Bean imbalanced	3	**0.915**	0.881	**0.917**	0.904	**0.913**	0.883	**0.917**	0.877	**0.914**	0.894
	5	**0.913**	0.872	**0.915**	0.889	**0.912**	0.883	**0.914**	0.893	**0.913**	0.893
	7	**0.911**	0.88	**0.915**	0.899	**0.911**	0.889	**0.91**	0.878	**0.914**	0.873
	9	**0.912**	0.877	**0.913**	0.891	**0.91**	0.875	**0.911**	0.875	**0.913**	0.889
	11	**0.912**	0.887	**0.914**	0.891	**0.908**	0.878	**0.911**	0.889	**0.911**	0.893
Dry Bean balanced	3	**0.915**	0.893	**0.918**	0.91	**0.913**	0.889	**0.916**	0.87	**0.913**	0.89
	5	**0.912**	0.876	**0.916**	0.9	**0.913**	0.884	**0.913**	0.859	**0.913**	0.88
	7	**0.911**	0.878	**0.915**	0.895	**0.913**	0.9	**0.912**	0.884	**0.911**	0.889
	9	**0.913**	0.881	**0.913**	0.89	**0.907**	0.881	**0.911**	0.87	**0.911**	0.881
	11	**0.911**	0.876	**0.911**	0.896	**0.913**	0.872	**0.912**	0.887	**0.909**	0.886

## Data Availability

Publicly available data sets were analyzed in this study. This data can be found here: [19].

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
