# Peer review of "Study on the Use of Artificially Generated Objects in the Process of Training MLP Neural Networks Based on Dispersed Data"

_entropy, 2023, doi:10.3390/e25050703_

Round 1

Reviewer 1 Report

The paper reports about an experiment for training MLP NN with artificially generated object.

The novelty with respect to other approaches referenced in the paper is that the objects are created locally (no risk of data privacy flaws) and then aggregated once and for all instead of iteratively as in state of the art approaches.

The experiment is reported with details of variants such that the experimented number of hidden layers, the different versions of data dispersion, and the like.

How the artificial objects are created is explained in the first part of the paper, and the results of all the experiments are reported in many tables where accuracy is shown.

The discussion then raises light about what variant of the experiments were more successful.

The paper is well written and the structure and the logical flow are easy to follow. I have only a remark related to the fact that Table 1 to 6 should be moved in the appendix as the occupy 6 pages in the middle of the paper and break its flow too much.

I find that the experiment per se has no particular deficiencies but, at the same time, I do neither see the innovation of it. 

One big thing is missing here and is the comparison with state of the art approaches that are referenced in the paper but not compared with the approach presented. For example: is the approach presented more successful in terms of accuracy than the iterative approach? What is the gain and the cost of the current approach with respect to the state of the art referenced in the paper? All these questions find no answers in the paper, but are of crucial importance, as this is a variant of previous experiments in the field of MLP NN training.

Although I may trust the goodness of the accuracy reported in the many tables of the paper, it is not clear whether the field has advanced with the approach presented or this is just an(other) isolated experiment in the field. I am afraid that we are in this second case, so I would not recommend publication in a Journal, unless a huge effort is made to better frame the approach presented in a comparative study or by means of a benchmark.

Author Response

Thank you sincerely for the comments and constructive advice. I believe these comments and advice will contribute much to the improvement of my paper. Changes are marked in the text. 

Table 1 to 6 were moved to the appendix. 

In the paper, an experimental comparison with an iterative approach modeled on federated learning is added. The approach and comparison are described in Section 3.5, Comparison of quality of classification of the proposed approach with an iterative approach modeled on federated learning. It is shown that we obtain better quality of classification when using the proposed approach. 

Reviewer 2 Report

This study concerns dispersed data stored in independent local tables with different sets of attributes. And with large amount of experiments, the paper gives out some interesting results. But the paper should be polished with more obvious statements.

Author Response

Thank you sincerely for the comments and constructive advice. 

A more readable sentence describing the obtained conclusions was added after each comparison of results. In addition, the conclusion sections were changed to be more readable.

Changes are marked in the text. 

Reviewer 3 Report

This paper deal with A Study on the use of artificially generated objects in the process of

training MLP neural networks based on dispersed data

1.      I suggest giving references or proof for each start Equations

2.      The paper it’s well written

Author Response

Thank you sincerely for the comments and constructive advice. 

Reference for each Equation was added.

Round 2

Reviewer 1 Report

The authors have addressed all of my main concerns about the work. the work has improved a lot, I have no other comments about it.